# Feasibility Study for the Development of a Low-Cost, Compact, and Fast Sensor for the Detection and Classification of Microplastics in the Marine Environment

**DOI:** 10.3390/s23084097

**Published:** 2023-04-19

**Authors:** Bruno Cocciaro, Silvia Merlino, Marco Bianucci, Claudio Casani, Vincenzo Palleschi

**Affiliations:** 1Consiglio Nazionale delle Ricerche—Istituto di Chimica dei Composti Organo-Metallici (CNR-ICCOM), U.O.S. di Pisa, Area della Ricerca del CNR, Via G. Moruzzi, 1, 56124 Pisa, Italy; 2Consiglio Nazionale delle Ricerche—Istituto di Scienze Marine (CNR-ISMAR), U.O.S. di Pozzuolo di Lerici, c/o Forte Santa Teresa—Loc. Pozzuolo di Lerici, 19032 Lerici, Italy; 3Dipartimento di Biologia, Università di Pisa, Via L. Ghini, 56124 Pisa, Italy

**Keywords:** microplastics, microplastic detection, marine environment, polyethylene, polypropylene, infrared reflectivity, low-cost sensors

## Abstract

The detection and classification of microplastics in the marine environment is a complex task that implies the use of delicate and expensive instrumentation. In this paper, we present a preliminary feasibility study for the development of a low-cost, compact microplastics sensor that could be mounted, in principle, on a float of drifters, for the monitoring of large marine surfaces. The preliminary results of the study indicate that a simple sensor equipped with three infrared-sensitive photodiodes can reach classification accuracies around 90% for the most-diffused floating microplastics in the marine environment (polyethylene and polypropylene).

## 1. Introduction

An emerging problem affecting the marine environment is the accumulation of Anthropogenic Marine Debris (AMDs), also called Marine Litter (ML), in the open sea [1,2,3] and emerged coastal areas [4,5,6,7,8,9,10], but also in deep water sediment [11,12,13], in polar areas [14,15], and in freshwater environments [16,17,18,19]. One performed survey confirmed that the percentage of “plastics” (including all artificial polymeric types) among the different categories of ML is very high, representing between 61% and 87% of AMDs [20,21]. The persistence of plastic materials and their slow degradation in the environment allows them to enter and accumulate in aquatic ecosystems, with possible harmful effects for the biota [22]. Macroplastics (2.5 cm^−1^ m), when subjected to chemical and physical stresses (photodegradation, mechanical shocks, etc.), fragment into smaller pieces [23,24,25], giving rise to second-generation microplastics. Added to these are “first-generation microplastics”, which are released into the aquatic ecosystem due to direct losses during their transfer and transport to/from industries. This second category includes microbeads (tiny microspheres used in cosmetics and other detergents, and nanoparticles used in industrial processes) and resin pellets, also called nurdles, a raw material used in molding and packaging to produce all kinds of plastic articles used in modern society [26,27,28]. Differences in the size of plastic pieces can lead to different types of dangerous effects on the environment and on the health of humans and animals, also due to chemicals and metals that might accumulate on their surface [29,30,31,32,33,34,35,36,37,38]. The proposed classification of microplastics (MPs) into size classes takes this fact into account [39].

It is therefore important, in line with protocol and standard monitoring strategies [40,41], to collect information on the spatial and temporal distribution of marine waste, especially on microplastics and their principal typology, to quantify their abundance and type, identify the areas of greatest accumulation, and possibly trace their origin and the contribution of different possible sources (rivers, tourism, etc.). 

The modalities of detecting and classifying marine microplastics is one of the most discussed issues of recent years. The need to be able to compare collected data has led the scientific community to converge on the adoption of some standard protocols that define how to implement collection and which parameters are the most interesting to classify. These include polymeric classification; several methods have been proposed and tested for the analysis of marine microplastics, based on their different spectral response in the infrared region of the spectrum (IR spectroscopy [42,43], micro-Raman [42,43,44,45,46], hyperspectral imaging [47]) or on their emission spectrum under the effect of a laser pulse [48]. All the above methods guarantee excellent discrimination of the main polymers that compose the Marine Litter, but the instrumentation needed for the analysis is typically restrained to laboratory applications. Microplastics are usually recovered from shorelines or sampled at sea according to established protocols, and their analysis is then carried out in the laboratory.

As part of the Tuscany Region’s MARTA (MultiplAtform smaRt drifTer—UAV—SAPR for marine investigation) Project, the authors investigated the feasibility of making a microplastic sensor to be mounted aboard a drifter fleet, so as to directly map the distribution of dispersed microplastics on the sea surface during their passive transport with currents.

The drifters typically used in the study of the sea currents are already equipped with temperature, pressure and salinity sensors, and in some cases, they can also collect information about marine micro-organisms by measuring the sea levels of O_2_, NO_3_ and particle organic carbon. Each drifter can travel several hundred or even thousands of kilometers from the time of release to their possible recovery (or at any rate, to the planned end of their mission), so each is capable of mapping a considerable area of water. For this reason, it would be extremely interesting to obtain, along with physical and microbiological information, a mapping of floating microplastics.

However, for reaching this goal, it is necessary to substantially reduce the size, complexity and, most importantly, the cost of the microplastics sensor. The latter condition is particularly critical for minimizing the economic losses from the likely loss of drifters (with attached sensors) after their release at sea. A reasonable target for the cost of each sensor can be around EUR 200/300. 

In Materials and Methods, we will describe which strategies have been adopted and which choices were implemented to pursue the goal we set. In Results, we will demonstrate the technical details of the prototype made and outline the first tests we carried out with it.

In Discussion and Conclusions, the initial results obtained will be discussed in light of the technical solutions adopted. The advantages and drawbacks of the sensor, future improvements, and possibilities for use in different contexts will be highlighted.

## 2. Materials and Methods

In our feasibility study for the realization of a robust, fast and cheap microplastics sensor, we considered the possibility of detecting and classifying the floating marine microplastics by exploiting their peculiar diffuse reflectance spectrum in the infrared region between 950 nm and 2600 nm. 

Several commercial instruments can operate in this region, with some of them also guaranteeing the size and robustness (but not the cost) requirements for being mounted on the drifters. Therefore, as a preliminary check, we analyzed a group of 218 plastic micro-pellets, recovered from several Italian beaches, with one of these miniaturized instruments (Neospectra from Si-Ware Systems, Cairo, Egypt). The instrument has a cost exceeding EUR 2000, which rules out its use on a fleet of floaters, but it is, nevertheless, very useful for the laboratory and in-situ characterization of microplastics. The Neospectra provides the diffuse reflectance spectrum of the sample between 1250 nm and 2250 nm. It operates at contact and acquires a spectrum in 2 s.

The 218 spectra were classified using the Graph Clustering method, which was introduced by the Pisa group in 2016 for the classification of samples based on their Laser-Induced Breakdown Spectroscopy spectra [49,50] and then also successfully applied to the classification of samples from their X-Ray Fluorescence spectra [51]. The graph is represented by an adjacency matrix whose (*i*, *j*) elements are given by the correlation between the *i* and *j* spectra:(1)$$A_{ij}=\frac{<S_{i}S_{j}>}{<S_{i}^{2}><S_{j}^{2}>}$$

The symbol $<S_{i}S_{j}>$ represents the scalar product between the two spectra *S_i_* and *S_j_*. The adjacency matrix *A_ij_* is square and symmetric (undirected graph). When the spectra are similar, as in the case of the marine pellets here studied, the elements of the adjacency matrix are typically very close to 1. To perform an effective classification, it is thus necessary to introduce a threshold value *th,* such that
(2)$$A_{ij}=0\;\;\;\;\;\;\;\;\;\;\;\;\;\;\;\;\;\;\;if\;A_{ij}<th$$

The value of *th* can be determined by maximizing the modularity [49] of the resulting graph, which is a measure of the compactness and separation of the clusters.

The graph corresponding to the 218 pellets here analyzed is shown in Figure 1.

The two main clusters resulting from the analysis contain 184 polyethylene (PE) and only 18 polypropylene (PP) pellets, respectively. The remaining 16 pellets were made of different materials, each one not correlated with other pellets; so, as they do not have connections, we do not show them in the graph. 

Having ascertained the possibility of discriminating from their reflectance properties the two materials most found among floating marine microplastics (PE and PP), despite the similarity of their spectra in the 950–2600 nm infrared region (see Figure 2), our next task became to find a way to obtain similar results while cutting by an order of magnitude the cost of the sensor used.

### Broaband InGaAs Sensors

We devised as a possible solution the use of broadband InGaAs sensors with different spectral sensitivity levels in the IR region, between 1250 nm and 2600 nm; they are described in Figure 3 (Laser Components GmbH, Olching, Germany).

The InGaAs photodiodes are alloys of InAs and GaAs deposited on an InP substrate. The energy bandgap of the alloy (and, consequently, the long wavelength cutoff of the photodiode) depends on the different proportions of the two components. The “standard” InGaAs alloy corresponds to a proportion of 53% InAs and 47% GaAs. This alloy has the same lattice constant of the InP substrate and a bandgap corresponding to 0.75 eV (at room temperature), which corresponds to a long wavelength cutoff around 1.7 μm. By increasing the InAs proportion in the alloy, the long wavelength cutoff increases, but a mismatch with the lattice constant of the substrate also occurs, leading to higher dark current and lower shunt impedance (slope of the current-voltage curve at V = 0) (see Table 1). 

To verify the feasibility of the use of InGaAs sensors for acquiring the diffuse reflectivity signal, we performed a classification study using as input the simulated signal that could have been obtained using the output of the five photodiodes, instead of the spectral data recovered with Neospectra instrument. 

Although the sensor is capable of distinguishing between different kinds of plastics (preliminary tests, not shown here, were performed on polystyrene (PS), polyvinyl chloride (PVC), polyethylene terephthalate (PET), polylactide (PLA) and polybutylene adipate terephthalate (PBAT)), in view of the specific application of the sensor to the detection of microplastic at open sea we focused on the possibility of detecting and classifying the two classes that represents the large majority of floating microplastics at sea, i.e., PE and PP. With the purpose of increasing the statistics (only 18 PP spectra were available), we applied to our data a bootstrapping method [52] that allowed us to increase the number of independent spectra up to 153, by averaging random couples of spectra to simulate a different spectrum. To avoid an imbalance between the 2 classes, 18 spectra were randomly selected among the 184 PE spectra available and the same bootstrapping method was applied also to them, and we obtained 153 PE spectra, i.e., the same number as in the PP case. 

The simulated signal of each sensor was obtained by convolving the IR spectrum with the sensor spectral response (see Figure 3). We used a simple Linear Discriminant Analysis (LDA) classifier [53], which gave an excellent accuracy in the classification, as shown in Figure 4.

The average True Positive Rate (TPR, percentage of the samples correctly classified) reaches 98% (min 96.7% for PE), which corresponds to an average False Negative Rate (FNR, percentage of the samples wrongly assigned to the class) of 2% (max 3.3% for PE).

To maintain low the cost of the instrument, we were forced to select only three among the five sensors available on the market. In this way, we could guarantee the cost of the prototype was around EUR 450, where 90% of this figure is given by the cost of the photodiodes, and the rest by lamp, focusing system, filter and electronics). Given the prices of the photodiodes on the market, keeping the cost of the single prototype under EUR 500 would probably guarantee a cost of EUR <300 per sensor for a small series production. The best combination of sensors was determined by looking for the highest classification rate that can be obtained while using the same strategy described above.

The results of the optimization are shown in Table 2.

It emerged that the best choice for the correct classification of PP and PE microplastics is the combination of the three InGaAs sensors with cuts at 1.7 μm, 1.9 μm and 2.2 μm. 

## 3. Results

### 3.1. Design and Realization of the Prototype Microplastic Sensor

Based on the above consideration, a conceptual scheme of the microplastics sensor is shown in Figure 5.

The illumination source is a small incandescence lamp (12 V, 3 W). The color temperature of the lamp is around 2500 K. According to the black body Wien formula, the wavelength corresponding to the maximum of the emission is
(3)$$\lambda_{max}=\frac{2898\;\mu m\cdot K}{T}$$

Therefore, the maximum of the spectral intensity is around 1.1 μm. According to the black body distribution: (4)$$I\left(\lambda \right)=\frac{2hc^{2}}{\lambda^{5}}\frac{1}{e^{\frac{hc}{\lambda k_{B}T}}-1}$$
at the wavelength of 2.2 μm, which corresponds to the cut of the third InGaAs photodiode, the lamp emission is about one-third of the intensity emitted at the maximum, thus guaranteeing a good signal up to that wavelength. The light is loosely focused through a cheap microscope objective on the water surface, where the microplastics are floating. The visible component of the light is cut using an IR-pass filter. The light reflected by the microplastics is finally collected by the InGaAs sensors, whose signals are amplified with a simple electronic circuit and instantly analyzed by the onboard microcomputer (we provide the logical scheme of the classification of the microplastics and the MatLab code used to train the classification model in Appendix A of the Appendix A).

The body of the sensor was designed (Figure 6) and 3D printed (Figure 7) using a biodegradable plastic material (polylactide—PLA). 

Unfortunately, because of the temporary unavailability of the InGaAs photodiode with a cut at 1.9 μm, for the test of the sensor we were forced to use the next-best configuration, with photodiodes at 1.7 μm, 2.2 μm and 2.6 μm. 

### 3.2. Microplastic Sensor Tests

At the time of writing this paper, we have started the first laboratory tests of the microplastic sensor. The pellets used for the preliminary feasibility test (Table 2) were not available anymore (they were subjected to destructive chemical analysis before the realization of the prototype, for studying the process of the accumulation of metals on their surface). For this reason, we tested the microplastic sensor (in the version made with the photodiodes at 1.7 μm, 2.2 μm and 2.6 μm) on HDPE and PP plastic objects collected from commercial packaging (8 bottle caps in HDPE, of different colors; 1 cap in PP; 5 PP lids; 1 PP transparent cup; and 1 PP plastic label), acquiring the signals of the 3 photodiodes in 103 different points for each sample. The total number of points samples was thus 1648 (824 on the 8 HDPE objects and 824 on the 8 PP objects). To reduce the effects produced by the physical differences (color and shape) of the samples, we normalized the signal of each photodiode to the total intensity measured on the three:(5)$$\begin{array}{c} S_{1.7}^{norm}=\frac{S_{1.7}}{\sqrt{S_{1.7}^{2}+S_{2.2}^{2}+S_{2.6}^{2}}}\\ S_{2.2}^{norm}=\frac{S_{2.2}}{\sqrt{S_{1.7}^{2}+S_{2.2}^{2}+S_{2.6}^{2}}}\\ S_{2.6}^{norm}=\frac{S_{2.6}}{\sqrt{S_{1.7}^{2}+S_{2.2}^{2}+S_{2.6}^{2}}} \end{array}$$

In Figure 8, the HDPE and PP objects analyzed are shown.

The results of the classification test obtained using the same Linear Discriminant Analysis (LDA) classifier used for the pellet spectra (see Section “Broaband InGaAs Sensors”, Figure 4) are shown in Figure 9.

We obtained an average TPR of 69.2%, corresponding to an average FNR of 30.8%. The minimum TPR was 64.7%, corresponding to a maximum FNR of 35.3%. 

The results are coherent with what was obtained on the pellet spectra with the same choice of sensors (average TPR of 75.8% and minimum TPR of 70.6%, see Table 2). It should be considered that the samples used for the test were very different in shape and color (see Figure 8). In comparison, the color and consistency of the pellets were much more similar between PE and PP samples, making their classification easier only on the basis of chemical composition.

It is important to note, in view of the use of the sensor on the floaters at sea, that the reflectivity of the marine water is negligible with respect to the signal coming from the plastics floating on the sea surface. Therefore, the results obtained in the laboratory are encouraging for the forthcoming tests of the sensor at sea.

## 4. Discussion and Conclusions

The results obtained so far, concerning the design of a microplastic detector with a low cost (a few hundred euros) but with sufficiently good performance, are definitely promising. Having used three InGaAs photodiodes instead of the five available was a compromise between high detection accuracy and low sensor cost. This last aspect, together with the fact that the sensor can be easily built, such as with a 3D printer, constitute the main value of the result obtained.

In addition, the use of biodegradable materials, among those available on the market today, is also an important factor to consider; it lessens the environmental impact that any leakage into the sea would have and thus makes it more environmentally friendly.

The choice of using LDA instead of Graph Clustering (or other more complex classification methods such as Artificial Neural Networks [53] or Self-Organizing Maps [54,55]) to evaluate the accuracy of classifying sample spectra was due to the need to implement an algorithm that could prospectively be used directly on board the marine drifters, on the same low-computational-power microcomputer used for real-time data acquisition. In any case, LDA provides excellent accuracy in the classification. The tests performed, from the first prototype of the sensor, on PE and PP samples showed good polymer discernment ability (Figure 9), with accuracy values comparable to those obtained for the previously analyzed pellets (Figure 4). This is an encouraging achievement that would justify further studies and tests, on pellets as well as on other types of microplastic. At the same time, initial tests are underway to impermeabilize the sensor in the best way without decreasing its efficiency, and to test its ability to function in the marine environment. This is a key point in the development of the instrument since we must guarantee its long lifetime in a harsh marine environment. In this direction, controlled tests are being implemented with the sensor mounted onboard mini-drifters (Figure 10). In these tests it has emerged that embedding the sensor in epoxy resin is probably the cheapest and most effective solution to impermeabilize and protect the sensor. During the next test, we will explore how the incorporation of the sensor into the resin works and whether it alters its optical properties.

We think it worthwhile to stress again the advantage of using consumer and readily available technologies for the microplastic sensor here presented; this fact allows for the direct involvement of common citizens in the Big Data acquisition process in this field, and raises awareness of the marine microplastic pollution problem and dimension. For example, the sensor could be realized in the frame of projects, such as the Italian “alternanza scuola lavoro” (“school-work internship”, now identified in Italy by the acronym PCTO), where numerous schools have already been successfully involved by the Authors in the collaborative development of simple tools for marine pollution monitoring [56]. Taking a cue from these approaches, we would also like to involve the community of the so called “makers ”, i.e., a large class of different people (technology enthusiasts, educators, thinkers, inventors, engineers, authors, artists, students, artisans, etc.), part of an international community that has a presence in over 100 countries, sharing the same spirit of the present project, which can be summarized in the two mottos “do it yourself” and, above all, “let us do it together”.

A low-cost, proximity microplastic sensor, such as the one presented here, lends itself to be used as an accessory, on pleasure boats, or on ferries/small boats that engage in passenger service in protected natural areas. In fact, these marine areas, such as, for example, the Cinque Terre Park or the Venice Lagoon area, are particularly vulnerable to pollution, including from microplastics, and are undergoing significant anthropogenic changes and are often near the outlets of important rivers. Providing incentives for the use of such a device in these vessels (or imposing it by regulation) would make a significant contribution to data collection, either for scientific use and to organize prevention, or, again, to increase the awareness of the problem among common citizens.

Finally, we are waiting for the possibility of using the best performing configuration for the three photodiodes (1.7 μm, 1.9 μm and 2.2 μm, Table 1), as soon as the 1.9 μm photodiode is available on the market again. This will certainly lead to a significant increase in performance (see Table 1) while maintaining the same final cost value for the entire sensor.

## Figures and Tables

**Figure 1 sensors-23-04097-f001:**
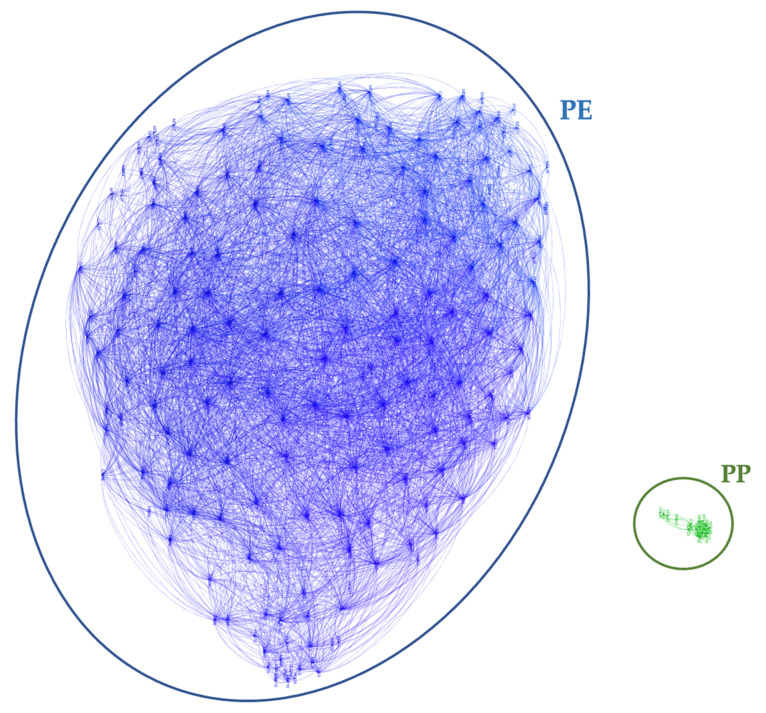
Clustering of the resin pellets according to their composition. The small cluster of green labels at the right corresponds to polypropylene (PP) pellets; the other cluster, far bigger, is of polyethylene (PE) pellets. The Graph Clustering method is able to discriminate, among the analyzed samples, the two types of polymers from their diffuse reflectance spectra. The graph was obtained by setting a correlation threshold of *th* = 0.9886.

**Figure 2 sensors-23-04097-f002:**
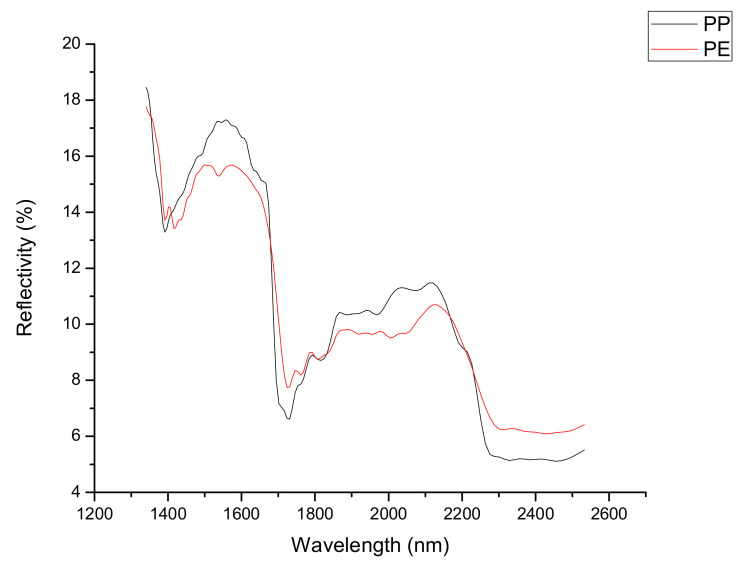
Average spectra of polypropylene (PP) and polyethylene (PE) pellets, taken with Neospectra instrument. The two spectra are very similar in almost the entire spectral region considered and show only slight differences around 1400 nm.

**Figure 3 sensors-23-04097-f003:**
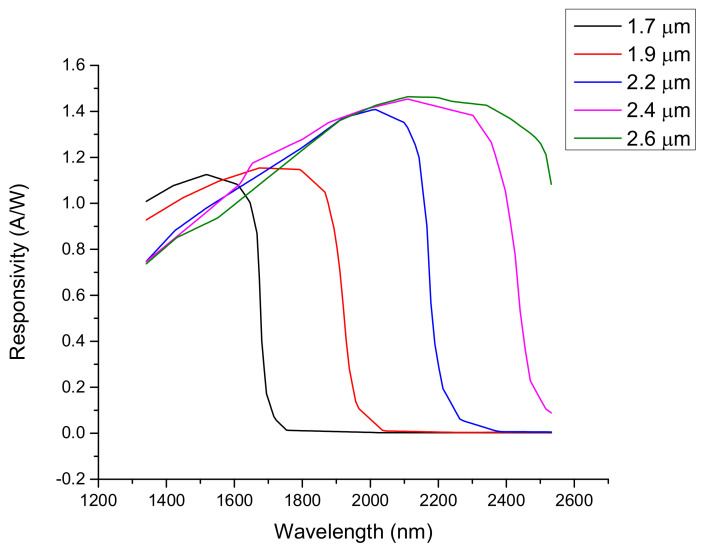
Typical spectral response of the InGaAs sensors available on the market. Each curve is labeled with the cut wavelength of the sensor.

**Figure 4 sensors-23-04097-f004:**
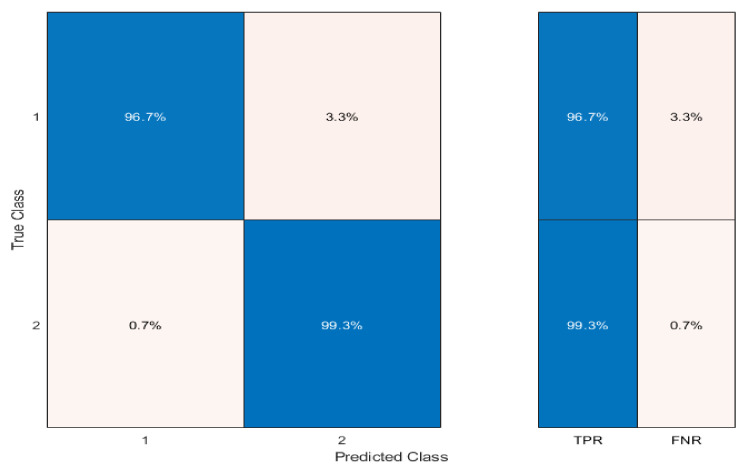
Confusion matrix for the classification of PP (class 1) and PE (class 2).

**Figure 5 sensors-23-04097-f005:**
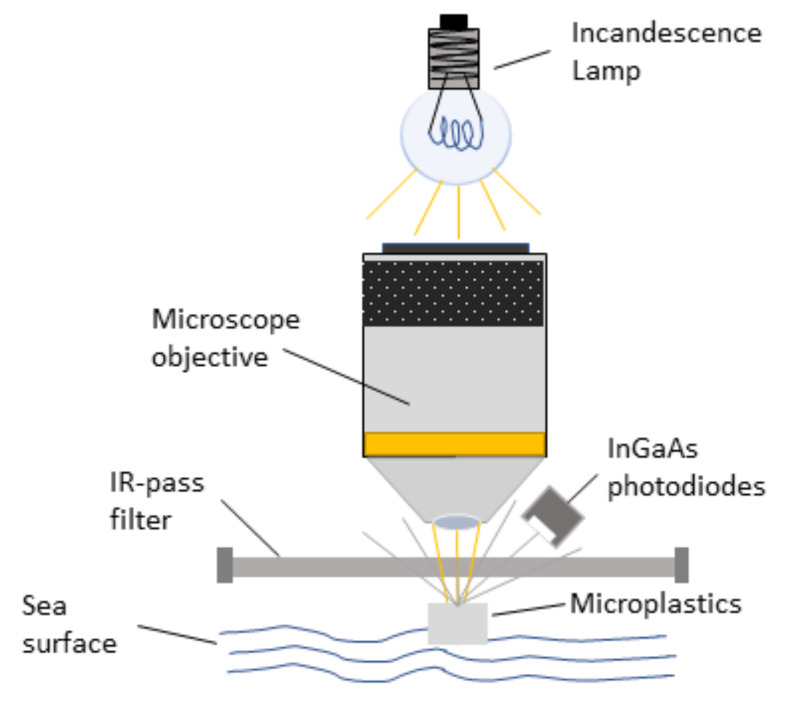
Schematic view of the microplastics sensor.

**Figure 6 sensors-23-04097-f006:**
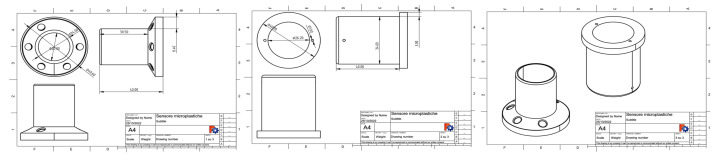
Schematic representation of the technical design of the microplastics sensor. Figures are not readable, but they are available from the authors on request.

**Figure 7 sensors-23-04097-f007:**
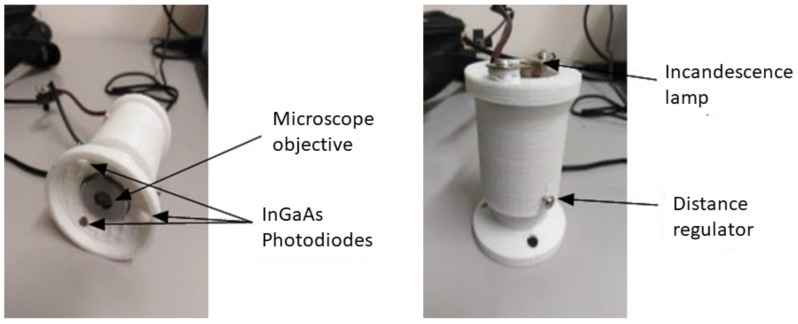
3D printed microplastics sensor (IR-pass filter not shown in the pictures).

**Figure 8 sensors-23-04097-f008:**
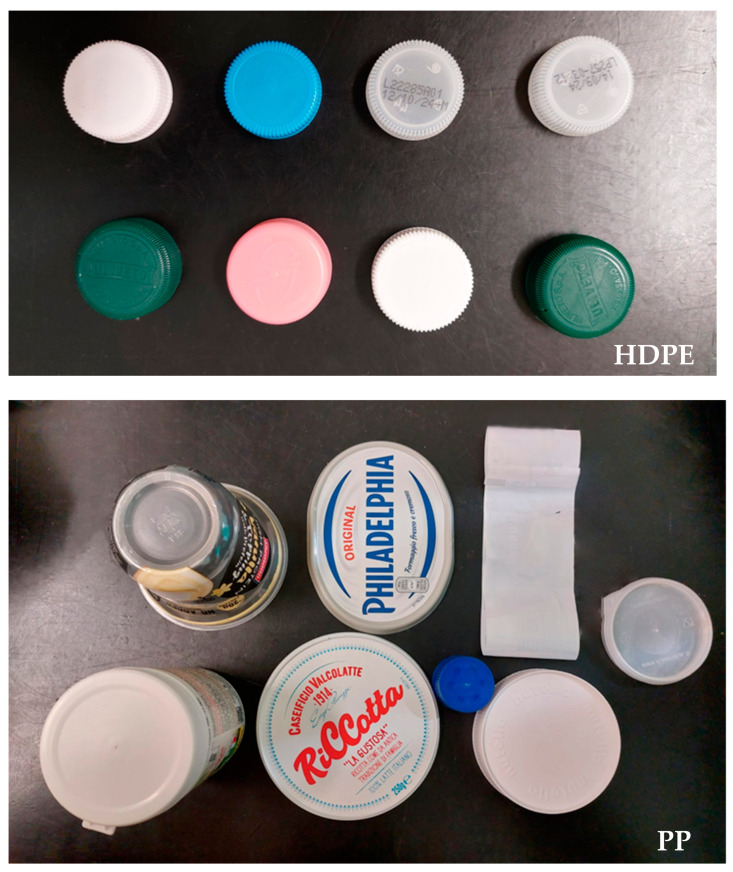
The plastic samples used for the test of the microplastic sensor.

**Figure 9 sensors-23-04097-f009:**
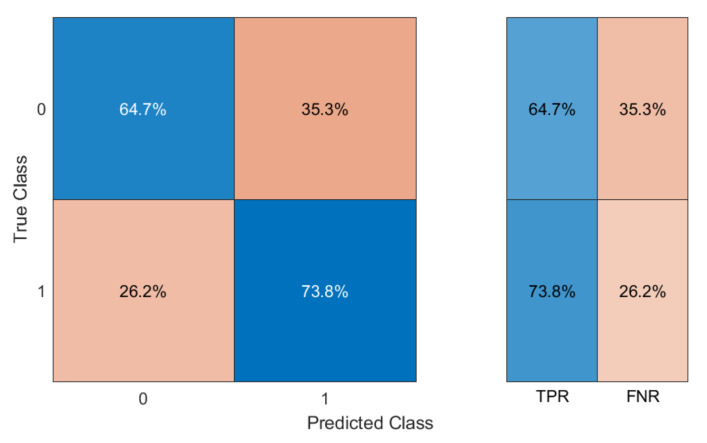
Confusion matrix for the classification of HDPE (class 0) and PP (class 1).

**Figure 10 sensors-23-04097-f010:**
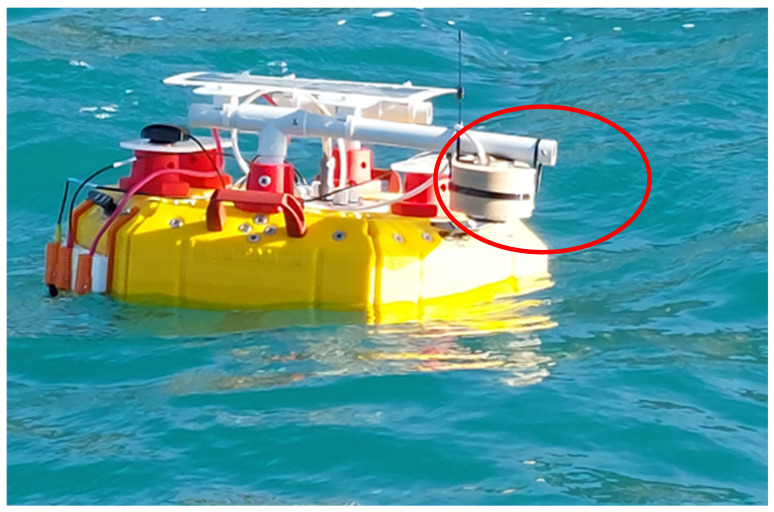
Impermeability test of the microplastic sensor (red circle) mounted onboard a smart-drifter.

**Table 1 sensors-23-04097-t001:** Typical dark current and shunt resistance of the different photodiodes (1 mm diameter).

Cutoff Wavelength (μm)	Dark Current @V_R_ = 0.25 V	Shunt Impedance @V_R_ = 10 mV
1.7	1 nA	100 MOhm
1.9	40 nA	1.6 MOhm
2.2	1 μA	78 KOhm
2.4	2.5 μA	20 KOhm
2.6	6 μA	5 KOhm

**Table 2 sensors-23-04097-t002:** Classification rates of different combinations of the 5 InGaAs available sensors.

Combination	Average TPR	Average FNR	Min TPR	Max FNR
(1.7, 1.9, 2.2)	90.8%	9.2%	90.8%	9.2%
(1.7, 2.2, 2.6)	75.8%	24.2%	70.6%	29.4%
(1.7, 1.9, 2.6)	61.1%	38.9%	58.8%	41.2%
(1.9, 2.4, 2.6)	58.8%	41.2%	56.2%	38.6%
(1.9, 2.2, 2.6)	58.5%	41.5%	56.2%	43.8%
(1.9, 2.2, 2.4)	56.5%	43.5%	55.6%	44.4%
(2.2, 2.4, 2.6)	54.2%	45.8%	53.6%	46.4%
(1.7, 2.4, 2.6)	51.3%	48.7%	49.0%	51.0%

## Data Availability

Data are available upon reasonable request.

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
