# Peer review of "Feasibility Study for the Development of a Low-Cost, Compact, and Fast Sensor for the Detection and Classification of Microplastics in the Marine Environment"

_sensors, 2023, doi:10.3390/s23084097_

Round 1

Reviewer 1 Report

Dear Authors,
Thank you for your work on a feasibility study to develop a low-cost, compact, fast sensor for detecting and classifying microplastics in marine environments.

The idea sounds good, and you provide an acceptable state-of-the-art, a good exposition of your vision and tests, and good evidence about its functionality.

I have only minor observations: figure 1 and the caption of figure 1 should stay on the same page, table 2 is the first of table 1, and table 2 has the line numbers on the left of the table (they should be on the right!).

I suggest you use LaTex next time.

Regards

Author Response

AUTHORS ANSWER:

First of all, we thank the reviewer for its fruitful comments. We revised some parts and improved the English as much as possible.

Below we list the changes made point by point and respond to their comments:

-we arrange figure 1 and its caption (now at lines from 138 to 142).

-The numbers of table 1 and table 2 have been changed, and line numbers corrected.

Reviewer 2 Report

It is necessary to share results from at least one test of the device in an uncontrolled environment with the most effective configuration before publishing the article.

 In the feasibility study, the cost of the developed device is not shown, only an estimate, and there is no comparison with other devices' costs. It is necessary to incorporate this information to maintain the integrity between what is presented and the article's title.

 They need to be more explicit, as there is still much information missing to fulfill the first paragraph of the Sensors template, which states: "The Materials and Methods should be described with sufficient details to allow others to replicate and build on the published results… make all materials, data, computer code, and protocols associated with the publication available to readers."

 There is a lack of information about the controlled experimental design and additional variables to consider, such as the refractive index of the water where the tests were carried out. Depending on this variable, the results can vary, as well as the depth at which the microplastics were found.

 Were the plastics on the surface of the water or on the table? At what depth or distance was the sensor tested? In Figure 10 of the conclusions, the microplastics are not visible.

 In Figure 5, they describe that the microplastics are on the surface of the ocean, and in Figure 10, there is a float where the mounted sensor can move the position of the microplastics. At what depth can the sensor detect microplastics?

Author Response

AUTHORS ANSWER:

First of all, we thank all the reviewers for their fruitful comments. We revised some parts and improved the English as much as possible.

Below we list the changes made point by point and respond to their comments:

point 1- In this manuscript, we present a concept paper: we have not done field tests (in the sea or lakes), and we will not do them until we can make the best sensor, the one with the three 1.7,1.9 and 2.2 micron photodiodes. We are waiting until we can purchase them to proceed. In the meantime, we continue with parallel tests, such as waterproofing tests.

point 2-  Paragraph added, from line 212 to line 216. The cost of the sensor with the photodiodes bought individually is around 450/500 Euro (bulb, lens, filter and electronics contribute a maximum of around 50 Euro, the rest is the photodiodes, total cost around 400 Euro). Let's say that if it were to be produced in small series, certainly, if not halved, the final cost would still fall below 300 Euro, which was our target from the beginning of the project. This cost has to be compared with the 2000 plus Euros of the portable FTIR (as specified from line 96 to line 98), which is the cheapest alternative that currently exists, and which, in any case, would have to be adapted for use at sea.

point 3-  In this regard, the paper includes information on the manufacturer of the photodiodes, and there are photos and technical drawings of the device. For everything else, we are more than willing to give any additional information on request from those interested.

point 4 - From line 303 to line 306: The measurement is made on floating plastics, and the sensor will be a few cm away from the water surface, whose reflectivity is negligible compared to the signal of microplastics (which are actually millimetric plastics in our case)

point 5 and 6 -  The tests reported were performed in the laboratory, placing the sensor at a working distance of about 5 cm from the sample. Other laboratory test (not reported in the paper) was done on a few samples floating on water; however, being the reflectivity of the water negligible, we preferred using larger samples to acquiring more easily the signals from different points on their surface.The sensor, as visible in Fig 10, is placed a few cm away from the water surface and would pick up the signal of the plastics floating on the water. We were not expecting to find microplastics in the test, that was focused on verifying the impermeability of the sensor in a real marine environment. However, even if some plastic fragment would have been present in the sea, it would have been very difficult to see them in the photo, as they are millimeter-sized.

Reviewer 3 Report

This study deals with the development of a low-cost sensor for detection of microplastics in marine environment. The results are well-presented and clearly organized. It focusses on polyethylene (PE) and polypropylene (PP) dispersed in marine environment. The manuscript is very interesting to achieve a sustainable world. Some Minor Revisions are suggested:

1. The authors should comment on possibility to detect other types of microplastics other than PE and PP, as described in the study.

2. What is the lifetime of the plastic sensor used in aggressive and extreme environment as in sea ?

3- Page 5. Line 157 - "1.7 mm" should be changed into "1.7 um"; in other terms micron and not millimeter.

After these revisions, the manuscript could be publishable.

Author Response

AUTHORS ANSWER:

First of all, we thank all the reviewers for their fruitful comments. We revised some parts and improved the English as much as possible.

Below we list the changes made point by point and respond to their comments:

point 1 - Considering the first comment, we added the following paragraph in the manuscript (from line 189 to 195): “Although the sensor is capable of distinguishing among different kinds of plastics (preliminary tests, not shown here, were performed on polystyrene (PS), polyvinyl chloride (PVC), polyethylene terephthalate (PET), polylactide (PLA) and polybutylene adipate terephthalate (PBAT)), in view of the specific application of the sensor to the detection of microplastic at open sea we focused on the possibility of detecting and classifying the two classes which represents the large majority of floating microplastics at sea, i.e. PE and PP.”

point 2 - The problem of making the sensor work in the harsh marine environment has been considered. We modify the following sentence (lines 339-347): “At the same time, initial tests are underway to impermeabilize the sensor in the best way, without decreasing its efficiency, and to test its ability to function in the marine environment. This is a key point in the development of the instrument, since we must guarantee the long lifetime in a harsh marine environment. In this direction, controlled tests are being implemented with the sensor mounted onboard mini-drifters (figure 10). In these tests it has emerged that embedding the sensor in epoxy resin is probably the cheaper and most effective solution to impermeabilize and protect the sensor. During the next test, we will explore how the incorporation of the sensor into the resin works and whether it alters its optical properties.”

point 3 - We apologize for the typo, we thank the reviewer for pointing it out to us. The error has now been corrected. (Line 173).

Round 2

Reviewer 2 Report

point 1- In this manuscript, we present a concept paper: we have not done field tests (in the sea or lakes), and we will not do them until we can make the best sensor, the one with the three 1.7,1.9 and 2.2 micron photodiodes. We are waiting until we can purchase them to proceed. In the meantime, we continue with parallel tests, such as waterproofing tests. Response.-  it´s ok

point 2-  Paragraph added, from line 212 to line 216. The cost of the sensor with the photodiodes bought individually is around 450/500 Euro (bulb, lens, filter and electronics contribute a maximum of around 50 Euro, the rest is the photodiodes, total cost around 400 Euro). Let's say that if it were to be produced in small series, certainly, if not halved, the final cost would still fall below 300 Euro, which was our target from the beginning of the project. This cost has to be compared with the 2000 plus Euros of the portable FTIR (as specified from line 96 to line 98), which is the cheapest alternative that currently exists, and which, in any case, would have to be adapted for use at sea.

Response.-  it´s ok

point 3-  In this regard, the paper includes information on the manufacturer of the photodiodes, and there are photos and technical drawings of the device. For everything else, we are more than willing to give any additional information on request from those interested.

Response.- You can include the flow diagram of the code developed and used, the computer code, and associated protocols.

point 4 - From line 303 to line 306: The measurement is made on floating plastics, and the sensor will be a few cm away from the water surface, whose reflectivity is negligible compared to the signal of microplastics (which are actually millimetric plastics in our case)

Response.- it´s ok

point 5 and 6 -  The tests reported were performed in the laboratory, placing the sensor at a working distance of about 5 cm from the sample. Other laboratory test (not reported in the paper) was done on a few samples floating on water; however, being the reflectivity of the water negligible, we preferred using larger samples to acquiring more easily the signals from different points on their surface.The sensor, as visible in Fig 10, is placed a few cm away from the water surface and would pick up the signal of the plastics floating on the water. We were not expecting to find microplastics in the test, that was focused on verifying the impermeability of the sensor in a real marine environment. However, even if some plastic fragment would have been present in the sea, it would have been very difficult to see them in the photo, as they are millimeter-sized.

    Response.- it´s ok

Author Response

At line 251 we added the following sentence:

"In Figure S1 of Supplementary Material the logical scheme of the classification of the microplastics and the MatLab code used to train the classification model)"

Moreover, we uploaded the document "Supplementary Material" reporting the computer code developed and the corresponding flow diagram